# The *Drosophila* miR-959–962 Cluster Members Repress Toll Signaling to Regulate Antibacterial Defense during Bacterial Infection

**DOI:** 10.3390/ijms22020886

**Published:** 2021-01-17

**Authors:** Ruimin Li, Xiaolong Yao, Hongjian Zhou, Ping Jin, Fei Ma

**Affiliations:** Laboratory for Comparative Genomics and Bioinformatics & Jiangsu Key Laboratory for Biodiversity and Biotechnology, College of Life Science, Nanjing Normal University, Nanjing 210046, China; 161201003@njnu.edu.cn (R.L.); 181202093@njnu.edu.cn (X.Y.); 181201007@njnu.edu.cn (H.Z.)

**Keywords:** miR-959–962 cluster, Toll pathway, *Toll*, *tube*, *dl*, *Drosophila melanogaster*

## Abstract

MicroRNAs (miRNAs) are a class of ~22 nt non-coding RNA molecules in metazoans capable of down-regulating target gene expression by binding to the complementary sites in the mRNA transcripts. Many individual miRNAs are implicated in a broad range of biological pathways, but functional characterization of miRNA clusters in concert is limited. Here, we report that miR-959–962 cluster (miR-959/960/961/962) can weaken *Drosophila* immune response to bacterial infection evidenced by the reduced expression of antimicrobial peptide *Drosomycin* (*Drs*) and short survival within 24 h upon infection. Each of the four miRNA members is confirmed to contribute to the reduced *Drs* expression and survival rate of *Drosophila*. Mechanically, RT-qPCR and Dual-luciferase reporter assay verify that *tube* and *dorsal* (*dl*) mRNAs, key components of Toll pathway, can simultaneously be targeted by miR-959 and miR-960, miR-961, and miR-962, respectively. Furthermore, miR-962 can even directly target to the 3′ untranslated region (UTR) of *Toll*. In addition, the dynamic expression pattern analysis in wild-type flies reveals that four miRNA members play important functions in *Drosophila* immune homeostasis restoration at the late stage of *Micrococcus luteus* (*M. luteus*) infection. Taken together, our results identify four miRNA members from miR-959–962 cluster as novel suppressors of Toll signaling and enrich the repertoire of immune-modulating miRNA in *Drosophila*.

## 1. Introduction

For the host, an appropriate immune response is essential to resist various pathogenic microorganisms and maintain the immune homeostasis. However, uncontrolled immune response would be detrimental to the host, eventually leading to the acute and chronic inflammatory disorders [1]. Determined by the speed and the specificity of the reaction, the innate and the adaptive immunities are vital for animal’s survival [2]. While invertebrates, such as *Drosophila melanogaster*, rely exclusively on innate immunity, as the first-line defense against microbial invaders [3,4]. The response of the flies to bacterial and fungal infections involves two main evolutionary conserved signaling pathways, Toll and immune deficiency (Imd) [5,6] which have been well-established. Upon systemic Gram-positive bacterial or fungal infection via septic injury, the Toll pathway is triggered, which involves extracellular proteolytic cascades activated by secreted recognition molecules (PGRP-SA, PGRP-SD, GNBP1, and GNBP3) [7,8,9,10,11]. Next, the transmembrane receptor Toll is activated and dimerized by the mature proteolytic product Spätzle [12,13,14,15], which subsequently causes the recruitment of three intracellular Death domain–containing proteins, MyD88, Tube, and Pelle [16,17,18]. Then the IκB homologue Cactus is phosphorylated and degraded by the proteasome, leading to the release of members of the nuclear factor NF-κB family (Dif or Dorsal) to translocate to the nucleus [19,20,21], and activation of genes encoding potent anti-fungal and anti-bacterial peptides, such as Drosomycin [7,22,23]. In addition, in response to Gram-negative bacterial infection, the Imd pathway is activated, eventually resulting that another *Drosophila* NF-κB family member Relish moves from the cytoplasm to the nucleus, and the expression of antimicrobial peptide (AMP) genes, such as *Diptericin* [22,24]. Therefore, the Toll and Imd immune pathways work together and constitute a robust defense system that protects *Drosophila* from invading pathogens [5].

The inactivation or overactivation of the immune response could lead to the damage of the normal tissue. Therefore, the activation and termination of the Toll pathway require the cooperation of various molecules at multiple stages to establish a complete immune regulatory system. At present, kinds of modulators have been identified to be involved in Toll pathway regulation. For example, five serine proteases (ModSP, Grass, Spirit, Spheroide, and Sphinx1/2), are considered as essential for host resistance to fungal and Gram-positive infection, which play a vital role in the extracellular proteolytic cascades linking the signaling recognition proteins and Spz [25,26]. In addition, a highly conserved protein Pellino, has shown to act as a positive regulator of Toll signaling by interacting with activated Pelle kinase [27]. Furthermore, in a genome-wide RNAi screens in S2 cells, G Protein-coupled receptor kinase 2 (Gprk2) was identified as a regulator of the Toll pathway [28], and the transcription factor DEAF-1 is confirmed to be required to induce Toll pathway target genes at or downstream of Dif/Dorsal [29]. Lastly, a feedback inhibitor is WntD, which reduces Toll activity by preventing translocation of Dorsal to the nucleus [30].

In addition to the above-mentioned protein regulatory factors, recently, growing evidences have exhibited that miRNA controls are a critical regulator in the immune response process via Toll pathway [31]. miRNAs could fine tune gene expression in diverse cellular and biological processes, through perfect or imperfect base-pairing to the 3′ UTR of the target mRNAs, resulting in cleavage or degradation of the target mRNAs or suppression of their translation [32,33]. For example, the transmembrane receptor Toll protein is a crucial factor connecting extracellular and intracellular signals, and it has been reported that miR-8 [34] and miR-958 [35] can target the 3′ UTR of its mRNA to negatively modulate the Toll pathway. Moreover, the nuclear translocation of the transcription factor Dif or Dorsal and its activation of AMP expression are an indispensable step of the Toll pathway response. miR-958 [35] and miR-317 [36] have been identified the direct binding with the 3′ UTR of *Dif-Ra/b/d* and *Dif-Rc* transcripts, respectively, while miR-8 targets to the *Dorsal* mRNA [34]. Last but not least, miR-310–313 family and miR-964 could directly target to the AMP gene *Drosomycin* to inhibit its expression [37,38]. Although several regulators involved in *Drosophila* Toll-mediated immune response have been identified, the restoration mechanism of *Drosophila* immune homeostasis is still largely unknown and needs for further research.

Especially, in our previous work, we found that the high-expression of four members of this miR-959–962 cluster could significantly down-regulate *Drs* expression via RNA-seq analysis and multiple genetic screening works [37]. Whether *Drosophila* miR-959–962 cluster members can synergistically repress Toll signaling to stop an overactive immune response, which is still not clear. In this study, we further investigated the regulatory mechanism of miR-959–962 cluster in the *Drosophila* immune response to bacterial infection. Each individual miRNA from the miR-959–962 cluster could reduce the survival rate of flies via inhibiting the expression of AMP *Drs*. Bioinformatics prediction and in vitro/in vivo experiments verified that four miRNA members (miR-959, miR-960, miR-961, and miR-962) could negatively regulated the Toll pathway in combination via directly targeting the 3′ UTR of *tube*, *dl*, or *Toll* mRNA. In addition, the dynamic expression pattern analysis demonstrated that four miRNA members were up-regulated at the late stage of *M. luteus* infection, revealing their important functions in the immune homeostasis restoration of *Drosophila*. Overall, our results have clarified that four miRNA members from miR-959–962 cluster are novel negative regulators in *Drosophila* Toll-mediated immune homeostasis restoration and their aberrant expression seriously influence *Drosophila* antibacterial defenses.

## 2. Results

### 2.1. The miR-959–962 Cluster Could Negatively Regulate Drosophila Toll-Related Immune Response

In order to assess the role of miR-959–962 cluster in *Drosophila* immune response, we first observed whether miR-959–962 cluster dysregulation would affect the resistance of *Drosophila* in response to lethal Gram-positive bacterial infection, *Enterococcus faecalis* (*E. faecalis*). As shown in Figure 1, the flies transiently overexpressed miR-959–962 cluster (*Gal80^ts^; Tub > miR-959*–*962*) under a temperature sensitive control and had a lower survival rate than the control flies (*Gal80^ts^; Tub-Gal4/+*) (Figure 1A). While the survival rate of the flies with miR-959–962 cluster knockout (*miR-959*–*962 KO*) was obviously increased, compared with the wild-type flies (*w^1118^*) (Figure 1B). We also confirmed the exact overexpression (Appendix A) and knockout (Appendix A) of each miRNA in the corresponding flies. These results suggest that the miR-959–962 cluster could weaken *Drosophila* antibacterial defense, implying a role for miR-959–962 cluster members in the negative regulation of Toll signaling.

To further confirm the effect of miR-959–962 cluster on the Toll pathway, we monitored the mRNA expression level of the AMP *Drs*, as the readout of Toll pathway activation, in the flies with miR-959–962 cluster overexpression and knockout before and after *M. luteus* infection. A significant reduction of *Drs* level was observed in miR-959–962 cluster overexpressing flies at 6 h and 12 h under bacterial challenge, compared with the corresponding control groups (Figure 1C). On the contrary, a higher expression level of *Drs* was detected in miR-959–962 KO flies than in wild-type controls (Figure 1D). Likewise, taking advantage of a Drosomycin–green fluorescent protein (GFP) reporter fly strain (Drs-GFP), we also observed that overexpression of miR-959–962 cluster inhibited the expression of Drs in live flies (94%) (Figure 1E), while the knock-out of miR-959–962 cluster increased the expression of Drs in live flies (39%) (Figure 1F). Our results suggest that four members of the miR-959–962 cluster may synergistically downregulate Toll signaling response to prevent overactivation of immune response and maintain *Drosophila* innate immune homeostasis.

### 2.2. Each Member from miR-959–962 Cluster Plays a Negative Regulatory Role in Drosophila Toll Pathway

To further explore the role of each miRNA individual from miR-959–962 cluster in *Drosophila* Toll pathway, transgenic lines high-expressing miR-959, miR-960, miR-961 or miR-962 separately (confirmed using RT-qPCR in Appendix A) were infected with *M. luteus*. The expression levels of *Drs* at 6 h and 12 h post-infection were also detected by RT-qPCR. Our result revealed that the *Drs* mRNA levels in the flies with miR-959 high-expression (*Gal80^ts^; Tub > miR-959*) (Figure 2A), miR-960 high-expression (*Gal80^ts^; Tub > miR-960*) (Figure 2B), miR-961 high-expression (*Gal80^ts^; Tub > miR-961*) (Figure 2C), or miR-962 high-expression (*Gal80^ts^; Tub > miR-962*) (Figure 2D) were significantly lower than that in the control flies (*Gal80^ts^; Tub-Gal4/+*) post-infection, respectively. Meanwhile, the corresponding flies carrying Drs-GFP reporter also suggested a decrease in the level of Drs in live flies upon infection (95%, 93%, 83% and 45%) (Figure 2E–G). In addition, the survival situation of individual miRNA high-expressing flies also were observed and recorded upon *E. faecalis* infection. Compared with the control groups, their survival ability was significantly reduced (Figure 3A–D). Taken together, these results indicate that each miRNA member from miR-959–962 cluster could inhibit the expression of AMP *Drs* and weaken the resistance to pathogen, to negatively fine tune *Drosophila* Toll signaling.

### 2.3. The Immune-Related Genes Are Potentially Targeted by miRNA Members from miR-959–962 Cluster In Vitro

In order to further determine how miR-959/960/961/962 regulates the Toll pathway, two algorithms, TargetScan and miRanda, were used to predict the potential target genes of miR-959, miR-960, miR-961, or miR-962. As described in the method, the intersection of two algorithms was acquired as the potential targets. Our results showed that miR-959 and miR-960 could bind with the 3′ UTR of *tube* mRNA, which is the crucial and indispensable effector molecule in the Toll pathway. Moreover, miR-961 and miR-962 had the base complementary pairs with the 3′ UTR of *dl* mRNA, a key transcription factor that activate the transcription of *AMP* genes. In addition, miR-962 also had a binding with the 3′ UTR of *Toll* mRNA, a transmembrane factor which transduce signals from extracellular to intracellular. The specific base complementary binding sites are shown in Figure 4A–C. These results suggest that miRNA members from miR-959–962 cluster may play fine-tuning functions at different levels of Toll signals transduction.

To evaluate the direct targeting relationship between miRNAs and targets, the 3′ UTR sequence of targets (*tube*, *dl*, and *Toll*) was respectively recombined to the downstream of the luciferase encoding sequence in the pAc 5.1 insect expression vector, as shown in the Figure 5A,D,G, and the Dual Luciferase Reporter Assay was carried out in *Drosophila* S2 Cell. The results showed that, compared with the pAc5.1 empty vector, both miR-959 and miR-960 could significantly reduce the activity of the luciferase reporter containing the 3′ UTR of *tube* (Figure 5B,C). The expression of luciferase reporter carrying with the 3′ UTR of *dl* could be markedly inhibited both miR-961 and miR-962 (Figure 5E,F). In addition, miR-962 could lower the luciferase activity of *Toll* 3′ UTR report plasmid (Figure 5H).

Furthermore, the target site mutation was performed in the 3′ UTR of *tube*, *dl*, and *Toll* as showed in the Figure 5A,D,G, in which the specific base mutation information is presented in Figure 4A–C. Dual Luciferase Reporter Assay found that the reporter activity of *tube*, *dl*, or *Toll* could be restored to the normal level in these cells with co-transfected the corresponding miRNA expression vector and 3′ UTR mutant reporters of *tube* (Figure 5B,C), *dl* (Figure 5E,F), or *Toll* (Figure 5H), identifying the reliability of the predicted target sites.

Taken together, our in-vitro results suggest that miR-962 could directly target the 3′ UTR of *Toll*, miR-959, and miR-960 target the 3′ UTR of *tube*, and miR-961 and miR-962 target the 3′ UTR of *dl*, indicating that different miRNA members from miR-959–962 cluster function on immunity by targeting different or identical immune-related genes.

### 2.4. The miR-959–962 Member Simultaneously or Serperately Target Key Components of Toll Pathway (Tube, dl, and Toll) In Vivo

To further confirm the reliability of predicted targets of miR-959, miR-960, miR-961 or miR-962 in *Drosophila*, we performed RT-qPCR analysis in vivo. Our results found that, compared with the control flies, the expression levels of *tube* mRNA in both miR-959 and miR-960 high-expressing flies were significantly down-regulated upon *M. luteus* infection (Figure 6A,B); Meanwhile the expression of *dl* mRNA in both miR-961 and miR-962 high-expressing flies also had a lower level than the controls (Figure 6C,D); In addition, the *Toll* mRNA level in miR-962 high-expressing flies was a significant reduction (Figure 6E). These suggest the negative correlations between these four miRNAs and corresponding targets in *Drosophila*.

### 2.5. Dynamic Expression Patterns of miR-959–962 Cluster Members in Wild-Type Flies after M. luteus or PBS Infection

To further explore the important role of this miR-959–962 cluster during Toll pathway response, we monitored the dynamic expression patterns of *Drs*, miR-959, miR-960, miR-961, and miR-962 in wild-type flies with *M. luteus* infection or PBS (control). Our results found that the levels of *Drs* in the *M. luteus* infected flies were significantly higher than the PBS-treated groups at 3, 6, 12, 24, 48 h, and peaked at 24 h after infection (Figure 7A). Subsequently, we detected the expression levels of miR-959–962 cluster members, respectively. We found that miR-959, miR-960, miR-961, and miR-962 (Figure 7B–E) were respectively significantly increased in the late stage of *M. luteus* infection. Taken together, we propose that the miR-959–962 cluster could play a crucial role in restoring *Drosophila* Toll immune homeostasis.

## 3. Discussion

Both the deficiency and overactivation of immune response are detrimental to *Drosophila*. Therefore, the persistence and intensity of the immune response needs to be strictly controlled to maintain the immune homeostasis [39]. At present, increasing evidences have demonstrated that some regulators, such as miRNAs, are involved in negatively regulating the immune signaling to prevent the over-activation of the immune response [34,40,41]. Recently, our group has performed a genome-wide miRNA screening to identify miRNAs regulating *Drosophila* Toll-mediated innate immune response, employing small-RNA seq and transgenic UAS-miRNA library [37]. Several potential miRNAs have been screened out, followed by in-depth exploration of their regulatory mechanism [35,36,37]. The current study found that the high-expression of miR-959–962 cluster in flies suppressed antibacterial defenses, evidenced by lower survival rate and a significant decrease of Drs expression in the presence of Gram-positive bacterial challenge.

Despite some reports on the contribution of single miRNA to *Drosophila* innate immune response have emerged, there are limited reports on how cluster of miRNAs work together. In this study, we demonstrated that each miRNA member of miR-959–962 cluster contributed to the suppression of antibacterial defense by targeting different components of Toll signaling pathway in a combinatory or separate manner, such as miR-959/miR-960 targeting *tube*, miR-961 repressing *dl*, and miR-962 targeting both *dl* and *Toll*. miRNA, perfectly complementary pairing with its target genes (Figure 4), leads to the cleavage and degradation of target mRNA to further block the expression of its protein [42,43]. Our results find that, in the flies with miR-959, miR-960, miR-961, or miR-962 high-expression, the corresponding target *tube*, *dl*, or *Toll* in mRNA level have a very significant decrease. Therefore, although no data are available, we believe that their protein levels is also sure to be significantly reduced.

MicroRNA clusters widely exist in metazoan genomes, employing with the diversity of their distribution [44]. Most of clustered miRNAs are located in polycistrons and co-expressed with adjacent miRNAs, causing the consistent expression patterns and levels [45,46]. On chromosome 2, the mature miR-959–962 cluster are transcribed from an intron of *CG31646* gene and the sequence of miR-963–964 cluster are within neighboring intron in *CG31646* gene. A previous study has showed that the six miRNA members from miR-959–964 cluster are probably encoded on a single transcription unit and showed a similar phase and amplitude [47]. Moreover, it has been indicated that the miR-959–964 cluster could inhibit *Drosophila* immune function against an attenuated strain of *Pseudomonas aeruginosa* [47]. In our study, of note that miR-960 may execute antibacterial defense only at late 12 h stage upon infection, while miR-959 may constantly repress the *Drs* expression at both 6 h and 12 h (Figure 2A,B and Figure 3A,B). Meanwhile, miR-961 may contribute more than miR-962 to repress antibacterial defense (Figure 2C,D; Figure 3C,D and Figure 5E,F). Therefore, we speculate that these miRNA of same cluster generated from the same transcripts with similar spatial-temporal expression pattern might have varied stability of half-lives, thus play a synergistic regulatory function on the Toll innate immunity via fine-tuning the different layers of Toll signal transduction (Figure 8).

Remarkably, in our work, the *Drs* expression and survival analysis shown in Figure 1 were performed under the background of miR-959–962 cluster high-expression or knockout flies (i.e., non-normal physiological conditions), whereas these dynamic expression patterns of four miRNA members of miR-959–962 cluster shown in this Figure 7 were performed under *M. luteus* infection and PBS treatment in the wild-type flies (i.e., normal physiological conditions). After the high expression of miR-959–962 cluster, *Drs* expression was down-regulated and the survival rate was reduced, implying that the miR-959–962 cluster played a negative regulator role in the Toll pathway (Figure 1). Thus we suggested that under the background of the high-expression of miR-959–962 cluster, miR-959/960/961/962 could inhibit the expression of immune-related target genes (e.g., *Toll*, *tube*, and *dl*) from the beginning of *M. luteus* infection, and lead to constant suppression of immune response in *Drosophila*. Therefore, compared with the control group, the flies with miR-959–962 cluster high-expression have an inadequate immune response, and its survival rate has been significantly reduced. Moreover, we analyzed the dynamic expression patterns of four miRNA members of the miR-959–962 cluster in wild-type flies to explore the endogenous role of miR-959/960/961/962 under normal physiological conditions, and we found that compared with PBS treatment groups, all four miRNA members were significantly increased in the late stage of *M. luteus* infection (Figure 7). Taken together, our results suggested that the miR-959–962 cluster plays a negative regulatory role in the later stage of immune response (Figure 8), i.e., in the early stage of *M. luteus* infection, the expression levels of *Drs* keep rising, and miR-959/960/961/962 is not up-regulated for avoiding the deficiency of immune response, but in the late stages of infection, in order to avoid the normal tissue damage caused by over-activation of immune response, miR-959/960/961/962 serve as negative regulators to down-regulate *Drs* expression to help *Drosophila* to restore to a new immune homeostasis.

In summary, our present studies have revealed the function of miR-959–962 cluster for inhibiting AMP expression and impairing antibacterial defenses. The functions and mechanisms of the four miRNAs from this cluster have also been identified, respectively. Therefore, our results not only identify a new function of miR-959–962 cluster, but also enrich the repertoire of Toll-related immune-modulating miRNA cluster in *Drosophila*.

## 4. Materials and Methods

### 4.1. Drosophila Stocks and Husbandry

Most flies were obtained from the Bloomington Drosophila Stock Center, including UAS-miR-959/960/961/962 (NO.60615), UAS-miR-959 (NO.60614), UAS-miR-961 (NO.41188), miR-959/960/961/962 KO (NO.58944), except UAS-miR-960 (F001954) and UAS-miR-962 (F001956) from FlyORF. *Drosophila* was raised on cornmeal-dextrose-yeast agar medium in a light-dark (12 h cycle) incubator at 25 °C and 60% humidity. To restrict miRNA overexpression to adulthood with tubulin-Gal80^ts^, the flies were reared and assayed in either 18 °C or 29 °C incubator.

### 4.2. Adult Immune Challenge

Control and miRNA mutant adult male flies, aged 2–4 days, were challenged by *Micrococcus luteus* (*M. luteus*), a widely used bacterial strain that activates the Toll-mediated immune response to induce the expression of the AMP *Drs*. Flies were firstly incubated at 29 °C for 24 h to activate the overexpression of miRNA. Septic injury was performed by pricking the thorax of the flies with a pulled glass capillary carrying *M. luteus* suspension mounted on a Nanoject apparatus (WPI, Sarasota, FL, UAS) [48], and then the flies were harvested at specified time points after treatment for RNA extraction and RT-qPCR. For the survival experiment, flies were infected with Gram-positive lethal bacteria, *Enterococcus faecalis* (*E. faecalis*), and their survival situation was monitored and recorded for 24 h post-infection [28].

### 4.3. Quantitative RT–PCR Analysis

Five adult flies per sample group were collected and isolated total RNA with TRIzol Reagent (Invitrogen, Waltham, MA, USA) according to the manufacturer’s protocol. RNA concentration and integrity was determined respectively by spectrophotometer and agarose gel separation. cDNA was synthesized using the HiScript^®^ II Q RT SuperMix for qPCR (Vazyme, Nanjing, China). Then quantitative PCR analysis was performed with the StepOnePlus Real-Time PCR System (Applied Biosystems, Foster City, CA, USA) using AceQ^®^qPCR SYBR Green Master Mix (High ROX Premixed) (Vazyme, Nanjing, China). Each experiment was performed in triplicate and the comparative cycle threshold was used to present a fold change for each specific mRNA/miRNA after normalizing to rp49/U6 snRNA levels. All primers we used in qPCR analyses are listed in Appendix A.

### 4.4. miRNA Targets Prediction

The mature sequences of miR-959/960/961/962 and 3′ UTR sequences of all genes in *Drosophila* were respectively acquired from miRBase and FlyBase database. Prediction analysis were carried out locally through employing TargetScan [49] and miRanda [50,51] software packages, applying their default parameters. To increase confidence and reduce false positive of the acquired miRNA-targets, the predicted results of TargetScan and miRanda were overlapped.

### 4.5. Recombinant Plasmids Generation

pAc5.1/V5-HisA insect expression vector was used for the recombinant plasmids construction. To introduce exogenous miR-959, miR-960, miR-961, and miR-962 in *Drosophila* S2 cells, the pre-miR-959, pre-miR-960, pre-miR-961, and pre-miR-962 sequence were amplified and cloned into pAc5.1/V5-HisA vector to generate pAc-miR-959, pAc-miR-960, pAc-miR-961, and pAc-miR-962 plasmids, respectively. The luciferase coding sequence was subcloned into pAc5.1/V5-HisA to generate pAc-luc. The 3′ UTR sequence of the *tube*, *dl*, and *Toll* transcript was respectively inserted to generate pAc-luc-tube 3′ UTR-wt, pAc-luc-dl 3′ UTR-wt and pAc-luc-Toll 3′ UTR-wt report plasmids, which were used to express the firefly luciferase. In addition, we also constructed 5 mutants of the above three report plasmids, respectively named pAc-luc-miR-959-tube 3′ UTR-mut, pAc-luc-miR-960-tube 3′ UTR-mut, pAc-luc-miR-961-dl 3′ UTR-mut, pAc-luc-miR-962-dl 3′ UTR-mut, and pAc-luc-miR-962-Toll 3′ UTR-mut. In the mutant plasmids, the original binding sites of corresponding miRNA was replaced with other bases without binding. All primers used are listed in Appendix A.

### 4.6. Cell Transfection and Luciferase Assays

*Drosophila* S2 cells were cultured in *Drosophila* standard medium (Gibco, Waltham, MA, USA) with 10% fetal bovine serum (Gibco, Waltham, MA, USA) at 28 °C. Before transfection experiment, cells were firstly seeded in 24-well plates (Corning, NY, USA) and cultured for 12 h. In each well, 385 ng miRNA expression plasmid, 100 ng pAc-luc-target 3′ UTR-wt (or pAc-luc-miRNA-target 3′ UTR-mut), and 15 ng pRL were co-transfected into cells by using X-tremeGENE HP DNA Transfection Reagent (Roche, Basel, Switzerland) according to manufacturer's instructions. The Renilla luciferase expressed by pRL was used as an internal reference. Dual luciferase assays were performed 48 h post-transfection with the Dual-Glo luciferase kit (Promega, Madison, WI, USA).

### 4.7. Data Processing and Statistical Analysis

Results from all experiments are presented as means ± SEM of the data. Statistical analyses were performed using two-tailed Student’s t-test, while statistical significance of survival experiment was calculated using the log-rank test (GraphPad Prism 7.04 software). For all statistical analysis, *p* < 0.05 was considered significant. All data significantly different from control values are marked with asterisks, * *p* < 0.05; ** *p* < 0.01; *** *p* < 0.001; and ns, no significance vs. the control.

## Figures and Tables

**Figure 1 ijms-22-00886-f001:**
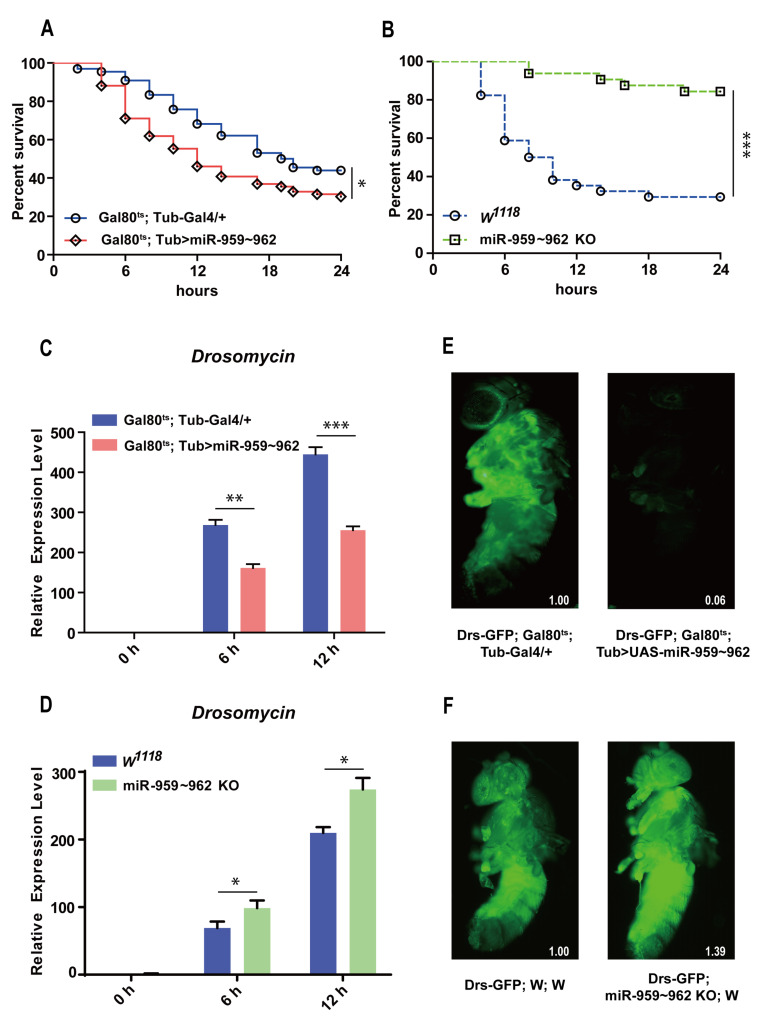
The miR-959–962 cluster negatively regulates *Drosophila* Toll-related immune response. (**A**) The changes of the survival rate were observed both in miR-959–962 cluster high-expressing flies (*Gal80^ts^; Tub > UAS-miR-959*–*962*) and the control (*Gal80^ts^; Tub-Gal4/+*) flies with *E. faecalis* infection. (**B**) The changes of the survival rate were observed both in miR-959–962 knock-out flies (*miR-959–962 KO*) and the control (*w^1118^*) flies upon *E. faecalis* infection. The expression levels of AMP *Drs* were examined in miR-959*–*962 cluster high-expressing flies (**C**) and miR-959*–*962 knock-out flies (**D**) at 0, 6 and 12 h upon *M. luteus* infection. (**E**) The green-fluorescent in miR-959*–*962 cluster high-expressing flies (*Drs-GFP; Gal80^ts^; Tub > UAS-miR-959–962*, right) and the controls (*Drs-GFP; Gal80^ts^; Tub-Gal4/+*, left) carrying with Drs-GFP reporter gene were observed under fluorescent microscope following infection with *M. luteus*. (**F**) The green-fluorescent in miR-959*–*962 cluster knock-out flies (*Drs-GFP; miR-959–962 KO; w*, right) and the controls (*Drs-GFP; w; w*, left) carrying with Drs-GFP reporter gene were observed under fluorescent microscope following infection with *M. luteus*. The levels of GFP were quantified using Image J software with the default parameters and their relative level values were marked in the bottom right corner of the image. (* *p* < 0.05; ** *p* < 0.01; *** *p* < 0.001).

**Figure 2 ijms-22-00886-f002:**
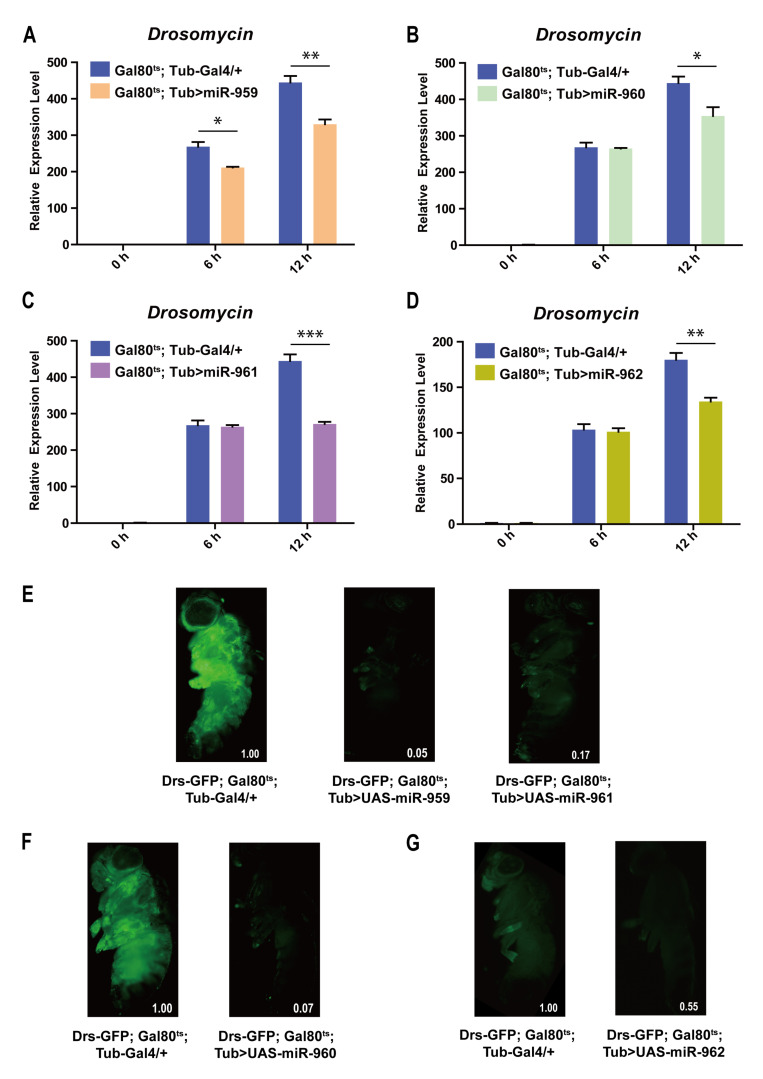
Each of miRNA member from miR-959–962 cluster inhibits the expression of *Drs* in *Drosophila* Toll immune response. The expression levels of AMP *Drs* were examined in miR-959 high-expressing flies (*Gal80^ts^; Tub > UAS-miR-959*) (**A**), miR-960 high-expressing flies (*Gal80^ts^; Tub > UAS-miR-960*) (**B**), miR-961 high-expressing flies (*Gal80^ts^; Tub > UAS-miR-961*) (**C**), and miR-962 high-expressing flies (*Gal80^ts^; Tub > UAS-miR-962*) (**D**), at 0, 6, and 12 h upon *M. luteus* infection. The green-fluorescent in miR-959 high-expressing flies (*Drs-GFP; Gal80^ts^; Tub > UAS-miR-959*, middle) (**E**), miR-960 high-expressing flies (*Drs-GFP;Gal80^ts^; Tub > UAS-miR-960*, right) (**F**), miR-961 high-expressing flies (*Drs-GFP;Gal80^ts^; Tub > UAS-miR-961*, right) (**E**) and miR-962 high-expressing flies (*Drs-GFP;Gal80^ts^; Tub > UAS-miR-962*, right) (**G**), and the controls (*Drs-GFP; Gal80^ts^; Tub-Gal4/+*, left) (**E**–**G**) carrying with Drs-GFP reporter gene were observed under fluorescent microscope at 12 h upon *M. luteus* infection. The levels of GFP were quantified using Image J software with the default parameters and their relative level values were marked in the bottom right corner of the image. (* *p* < 0.05; ** *p* < 0.01; *** *p* < 0.001).

**Figure 3 ijms-22-00886-f003:**
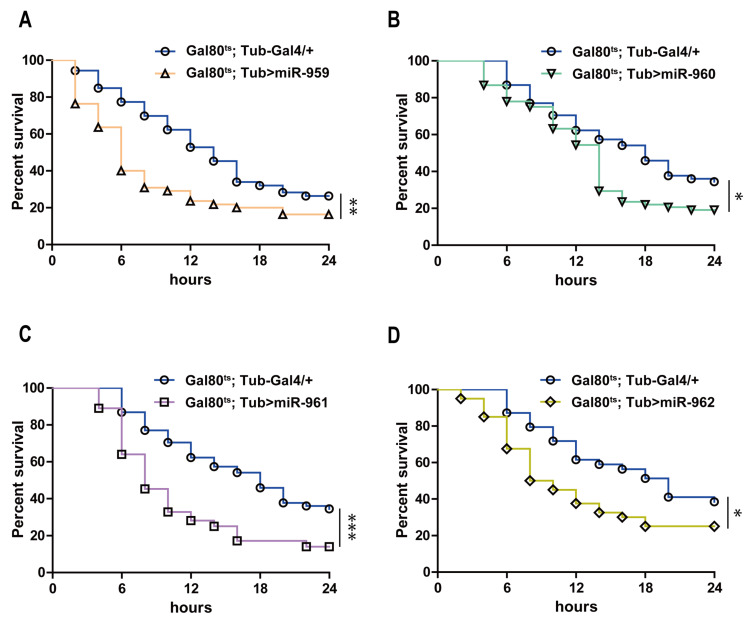
Each of miRNA member from miR-959–962 clusters influences the survival of *Drosophila*. The changes of the survival rate were observed in miR-959 high-expressing flies (**A**), miR-960 high-expressing flies (**B**), miR-961 high-expressing flies (**C**) and miR-962 high-expressing flies (**D**), as well as the control flies upon *E. faecalis* infection. (* *p* < 0.05; ** *p* < 0.01; *** *p* < 0.001).

**Figure 4 ijms-22-00886-f004:**
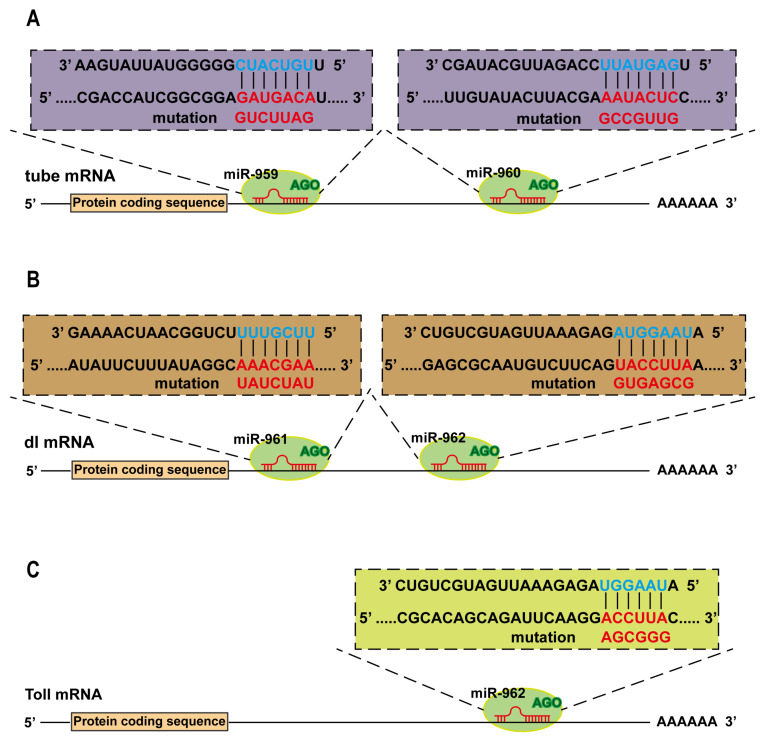
The target genes of four miRNA members from miR-959–962 clusters were predicted. The potential binding sites of miR-959, miR-960, miR-961, and miR-962 in the 3′ UTR of *tube* (**A**), *dl* (**B**) and *Toll* (**C**) were present, respectively. The point mutations (red) at the 3′ UTR target sites base pairing to the seed sequence of corresponding miRNA (blue) were performed.

**Figure 5 ijms-22-00886-f005:**
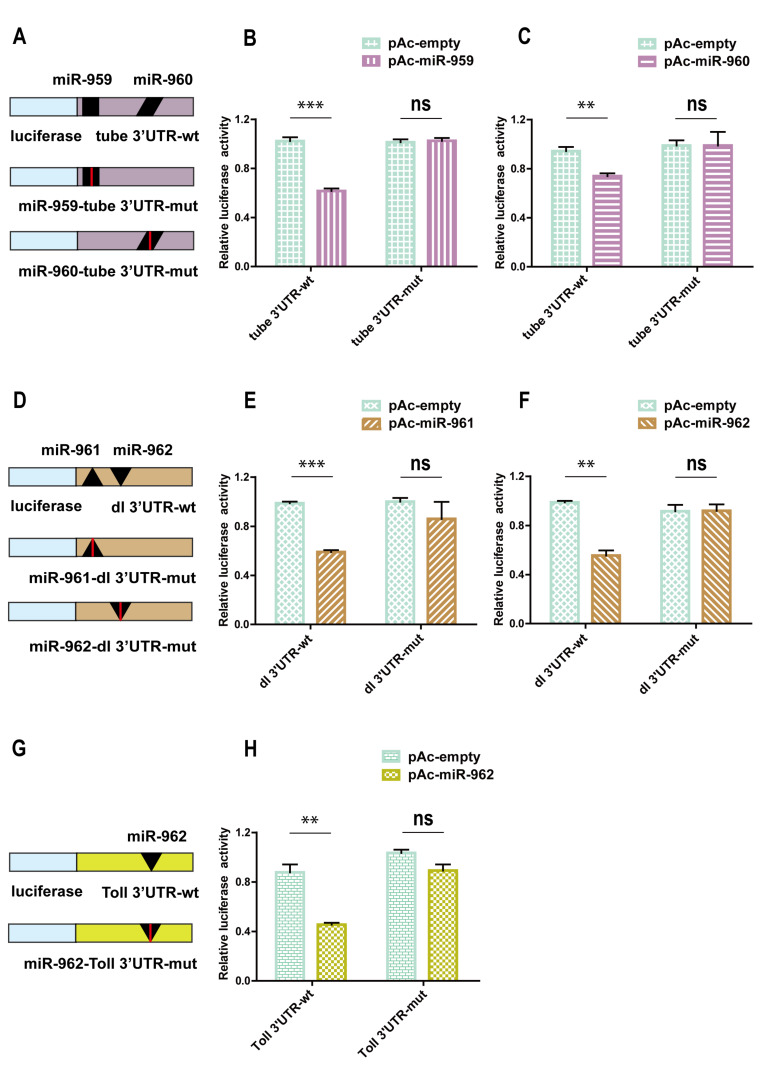
The direct bind between four miRNA members from miR-959–962 cluster and its target genes were confirmed by Dual luciferase reporter assay *in vitro*. (**A**,**D**,**G**) The schematic diagrams of construction of targets 3′ UTR and 3′ UTR mutation luciferase reporter plasmids were presented. After co-transfected with miRNA expression plasmid, the corresponding luciferase activity of the report plasmids without or with mutation sites was determined in *Drosophila* S2 cell on a Dual luciferase assay (**B**,**C**,**E**,**F**,**H**). (** *p* < 0.01; *** *p* < 0.001; and ns, no significance vs. the control).

**Figure 6 ijms-22-00886-f006:**
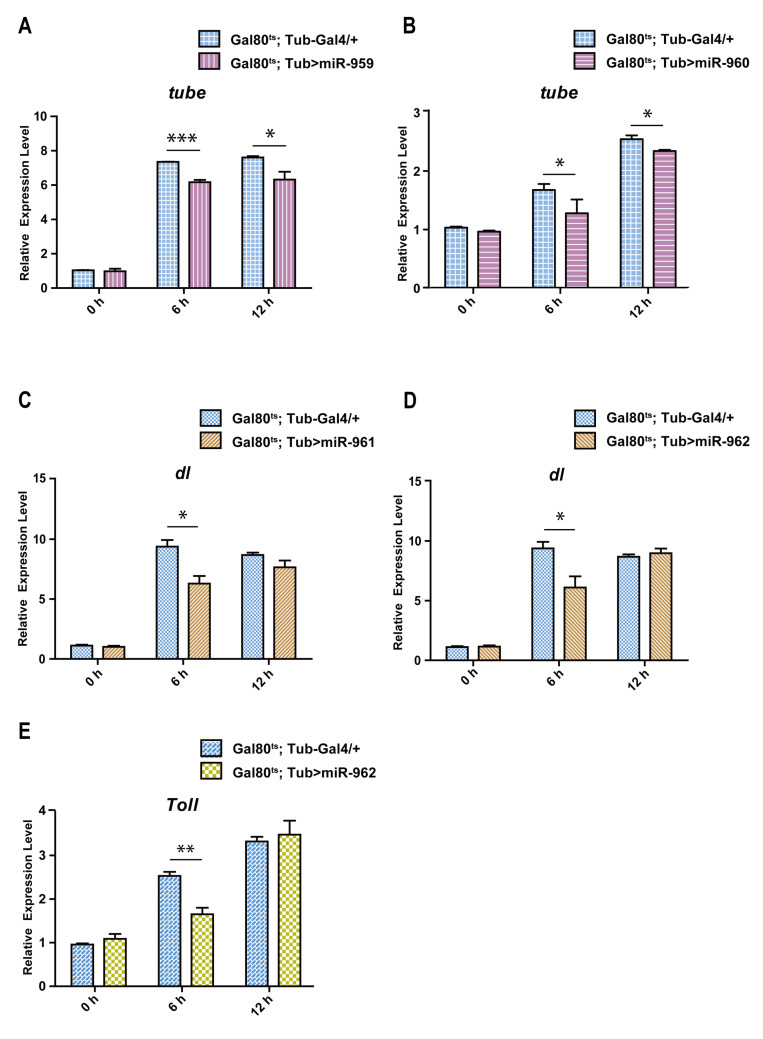
Four miRNA members from miR-959–962 cluster inhibit the expression of its target genes in vivo. The expression levels of *tube* were respectively tested in miR-959 high-expressing (**A**) and miR-960 high-expressing flies (**B**). (**C**,**D**) The expression levels of *dl* in miR-961 high-expressing and miR-962 high-expressing flies were respectively tested. (**E**) The expression level of *Toll* in miR-962 high-expressing flies was detected. (* *p* < 0.05; ** *p* < 0.01; *** *p* < 0.001).

**Figure 7 ijms-22-00886-f007:**
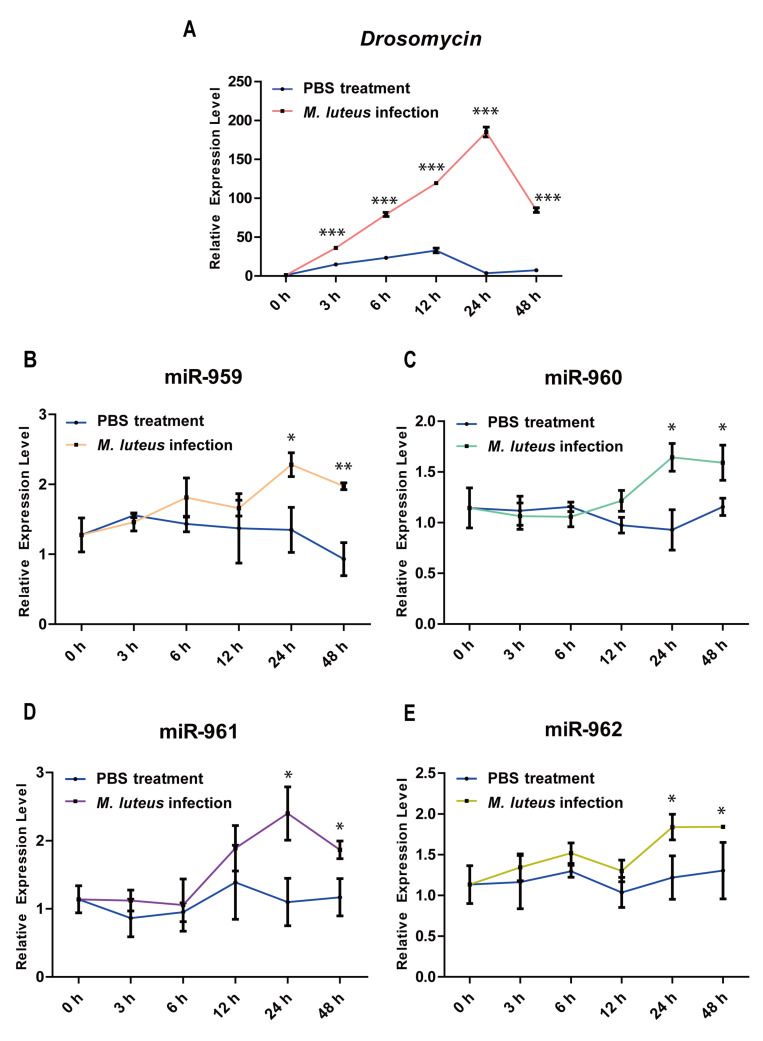
The temporal expression patterns of four miRNAs in the wild-type flies prior to and following *M. luteus* infection. The dynamic expression changes of *Drs* (**A**), miR-959 (**B**), miR-960 (**C**), miR-961 (**D**), miR-962 (**E**) at six time-points (0, 3, 6, 12, 24, and 48 h) prior to and following *M. luteus* or PBS infection, respectively. (* *p* < 0.05; ** *p* < 0.01; *** *p* < 0.001; and ns, no significance vs. the control).

**Figure 8 ijms-22-00886-f008:**
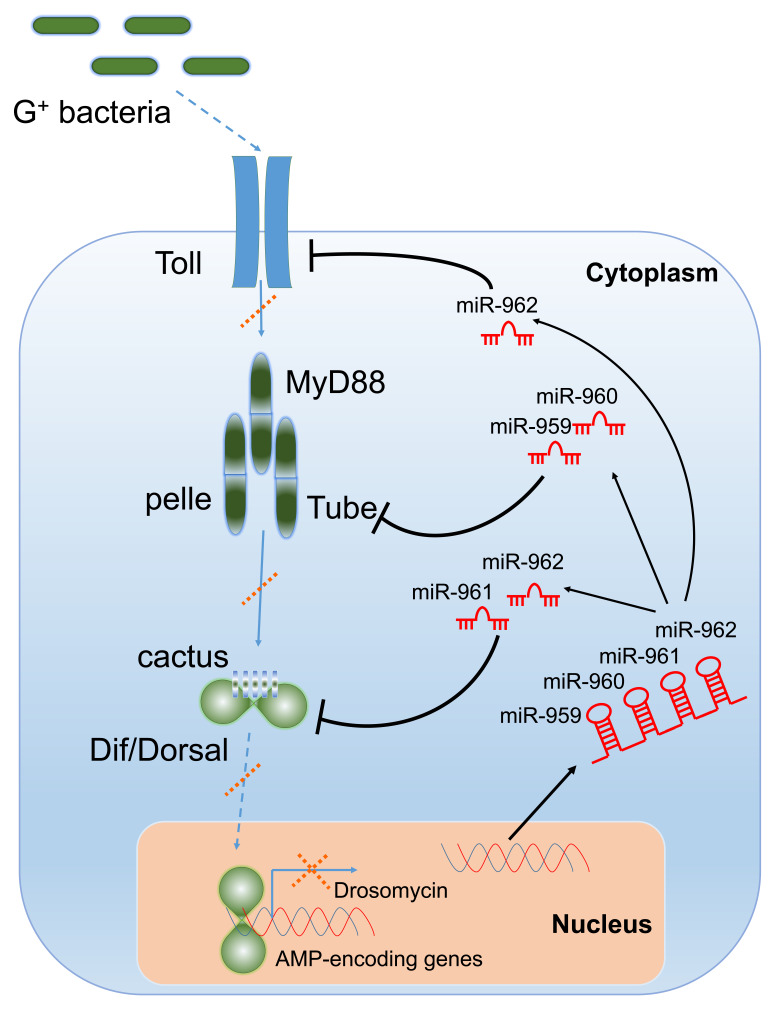
A proposed model. Our results suggested a model in which the miR-959–962 cluster members (red) play a synergistic regulatory function on the Toll innate immunity via fine-tuning the different layers of Toll signal transduction. MiR-959 and miR-960 target the 3′ UTR of *tube*; miR-961 and miR-962 target the 3′ UTR of *dl*; and miR-962 also target the 3′ UTR of *Toll*.

## Data Availability

All relevant data are within the manuscript and its Supporting Materials.

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
