# Peer review of "The *Drosophila* miR-959–962 Cluster Members Repress Toll Signaling to Regulate Antibacterial Defense during Bacterial Infection"

_ijms, 2021, doi:10.3390/ijms22020886_

Round 1

Reviewer 1 Report

The authors have addressed my concerns and the paper is now ready for publication in its current form.

Author Response

Thank you very much for the affirmation of our revision.

Reviewer 2 Report

Major concerns

This revised manuscript from Li et al. clearly verified the molecular mechanism of miR-959~962 inhibiting Toll signaling pathway in Drosophila. However, the story looked like the authors already knew that miR-959~962 will suppress Toll signaling. So many miRNAs also had the potential ability in regulation of Toll signaling. The most important key point is why authors pick up the miR-959~962 cluster to investigate its’ role in regulation of Toll signaling? I recommended that the authors should provide rational hypothesis for why picked up and evaluated the miR-959~962 cluster in targeting immune response prior to present the functional assay. The figures and story should rearrange.

In fig 7 and 8, the authors aimed to claim that miR-959~962 cluster served as negative regulators to restore immune homeostasis. This conclusion seemed to conflict with fig 1. Do authors want to told audiences When Toll signaling in innate immune cells stimulated by infection, the mi959~962 cluster simultaneously over-expressed to regulate immune responses, and hosts may have shorter survival rate? The correlation between survival rate and physiologic function of miR-959~962 should interpret clearly.

Author Response

This manuscript is a resubmission of an earlier submission. The following is a list of the peer review reports and author responses from that submission.

Round 1

Reviewer 1 Report

In this manuscript, Xialong and colleagues study the regulatory roles of the miR-959-962 miRNA cluster in the Toll signaling and its influence in the immune response to bacterial infection, by using Drosophila melanogaster and S2 cells as models systems. They find that manipulation of these miRNAs leads to changes in Toll signaling and fly viability after bacterial infection. They identify specific members of this pathway as predicted targets of the miRNA cluster. They perform luciferase assay to validate the functionality of those predicted sites and observe that miRNA manipulation in vivo leads to changes in the mRNA of those predicted target genes. In light of those observations the authors suggest that this cluster dampens Toll activation in antibacterial response. The manuscript is potentially interesting but some issues need to be addressed before this work is ready for publication.

Importantly, mutants of this cluster seem to do better than control flies (Fig 1B). Therefore, why flies have maintained this miRNA cluster? Interestingly, different reports have demonstrate that miRNAs are, in many cases, embedded in feedback/feedforward loops to control different cellular responses aimed at increase robustness. In this work, it would be central to show whether the Toll pathway is controlling the expression of this miRNA cluster. If this were the case, this would provide evidence of yet another case where miRNAs respond to signaling activation by dampening the levels of pathway activity. This experiment would give a biological role that, in its current state, this MS is lacking.

The authors use ubiquitous Gal4 drivers to manipulate the expression of the miRNAs analyzed. This approach does not allow to identify the tissue in which this cluster controls the Toll pathway. It would be crucial to determine the specific cell type in which this miRNAs are regulating the Toll cascade.

Other specific comments below:

  • Fig 1A-B and Fig 3 show the consequences of miR-959-962 miRNA cluster manipulation after infection. The authors should show whether those changes in viability take place in absence of bacterial infection. The miRNA cluster could regulate other targets that could account for the changes in viability observed. These controls are required to determine if the effect is specific or not to a context of bacterial infection.

  • Fig 1C,E show changes in Drs upon miRNA cluster over-expression. The authors should also perform those analyses in a miRNA cluster KO background. This would show whether the results observed are physiologically relevant.

  • The authors claim in the text: “These above results indicate miR-959~962 cluster undermine Drosophila antibacterial defenses by down-regulating Toll-mediated immune pathway.” The results obtained are not sufficient to make that claim. To reach that conclusion, the authors should restore the Toll activity levels and show that this rescues the phenotype created by the manipulation of the miRNA cluster. Without those functional approach the authors can´t present such conclusions.

Minor concerns:

  • The manuscript contains some grammatical mistakes that need to be corrected. Eg, Fig 17 in the abstract, “Each of the four miRNA member was confirmed…” should read ““Each of the four miRNA members was confirmed…”; Fig legend 1, “The miR-959~962 cluster negatively regulate Drosophila…” should read “The miR-959~962 cluster negatively regulates Drosophila…”, etc.

  • The font sizes in Fig 4 are very small and it is difficult to read. The authors should make a deep revision of the Fig to make all the information included there readable.

Reviewer 2 Report

Overall

I have carefully read the manuscript – the data is clearly presented and the English is fairly good, with changes suggested below. The data in Figs 1,2 and 6 could be improved as described below.

Critique

All GFP panels in fig 1,2 should be labeled directly and separated – difficult to see as separate data for comparison purposes

Fig 6: For me, it remains unclear and goes unremarked that mir-162 effects on dl and Toll levels are less at 12h compared to 6H. These data arguing for specificity of each mir for corresponding targets could be strengthened by testing cross effects, for example showing that mir-159 affects tube and not dorsal expression as would be expected by algorithm evaluation.

Minor changes

Line 31 “immunities” rather

Line 33 incomplete sentence

Line 41 “release of members” rather

Line 52 “has been emerging as” – change to “are”

Line 74-75 run on sentence

Line 77 incomplete sentence

Line 78 “the exact each” better structure

Line 114 “also suggested” here and elsewhere use software to ascribe a number to the value of decreased Drs-GFP

Line 180 delete “which”

Line 224 “separate” rather